# Food Insecurity Is Associated with Diet Quality in Pregnancy: A Cross-Sectional Study

**DOI:** 10.3390/nu16091319

**Published:** 2024-04-28

**Authors:** Bree Whiteoak, Samantha L. Dawson, Leonie Callaway, Susan de Jersey, Victoria Eley, Joanna Evans, Alka Kothari, Severine Navarro, Danielle Gallegos

**Affiliations:** 1School of Exercise and Nutrition Sciences, Faculty of Health, Queensland University of Technology (QUT), 149 Victoria Park Road, Kelvin Grove, QLD 4059, Australia; danielle.gallegos@qut.edu.au; 2Centre for Childhood Nutrition Research, Faculty of Health, Queensland University of Technology (QUT), 62 Graham Street, South Brisbane, QLD 4101, Australia; severine.navarro@qimrberghofer.edu.au; 3QIMR Berghofer Medical Research Institute, 300 Herston Rd., Herston, QLD 4006, Australia; 4Food & Mood Centre, IMPACT—The Institute for Mental and Physical Health and Clinical Translation, School of Medicine, Barwon Health, Deakin University, Geelong, VIC 3220, Australia; samantha.dawson1@deakin.edu.au; 5Women’s and Newborns Services, Royal Brisbane and Women’s Hospital, Herston, QLD 4006, Australia; leonie.callaway@health.qld.gov.au; 6Faculty of Medicine, The University of Queensland, 288 Herston Rd., Herston, QLD 4006, Australia; v.eley@uq.edu.au (V.E.); alka.kothari@uq.edu.au (A.K.); 7Department of Dietetics and Foodservices, Royal Brisbane and Women’s Hospital, Herston, QLD 4006, Australia; susan.dejersey@health.qld.gov.au; 8Centre for Health Services Research, Faculty of Medicine, The University of Queensland, 288 Herston Rd., Herston, QLD 4006, Australia; 9Department of Anaesthesia and Perioperative Medicine, Royal Brisbane and Women’s Hospital, Herston, QLD 4006, Australia; 10Maternity Services, Caboolture Hospital, McKean Street, Caboolture, QLD 4510, Australia; joanna.evans@health.qld.gov.au; 11Redcliffe Hospital, Anzac Avenue, Redcliffe, QLD 4020, Australia

**Keywords:** food insecurity, food security, pregnancy, diet quality, dietary intake

## Abstract

Household food insecurity (HFI) and poorer prenatal diet quality are both associated with adverse perinatal outcomes. However, research assessing the relationship between HFI and diet quality in pregnancy is limited. A cross-sectional online survey was conducted to examine the relationship between HFI and diet quality among 1540 pregnant women in Australia. Multiple linear regression models were used to examine the associations between HFI severity (marginal, low, and very low food security compared to high food security) and diet quality and variety, adjusting for age, education, equivalised household income, and relationship status. Logistic regression models were used to assess the associations between HFI and the odds of meeting fruit and vegetable recommendations, adjusting for education. Marginal, low, and very low food security were associated with poorer prenatal diet quality (adj β = −1.9, −3.6, and −5.3, respectively; *p* < 0.05), and very low food security was associated with a lower dietary variety (adj β = −0.5, *p* < 0.001). An association was also observed between HFI and lower odds of meeting fruit (adjusted odds ratio [AOR]: 0.61, 95% CI: 0.49–0.76, *p* < 0.001) and vegetable (AOR: 0.40, 95% CI: 0.19–0.84, *p* = 0.016) recommendations. Future research should seek to understand what policy and service system changes are required to reduce diet-related disparities in pregnancy.

## 1. Introduction

Nutrition in pregnancy is of vital importance to short- and long-term maternal and child health [1,2]. Higher prenatal diet quality and healthier dietary patterns have been associated with a reduced risk of adverse perinatal outcomes, including maternal depression, hypertensive disorders of pregnancy, preterm birth, and low birthweight [3]. These can all have long-lasting effects on health and development [2,4,5,6]. Diet quality is therefore increasingly the focus of research aiming to inform diet-related public heath recommendations [3,7].

While a high proportion of women agree that eating well during pregnancy is important [8,9] and report high levels of motivation to do so [10], prenatal diet quality is generally poor in Australia [10,11,12,13] and other high-income countries [14,15,16]. It does, however, follow a social gradient, with better prenatal diet quality frequently observed among those with higher educational attainment and income levels [11,14,15,16,17,18]. 

Lower educational attainment and lower incomes, in combination with other social factors, are associated with household food insecurity [19,20]. This occurs when there is limited or uncertain access to enough safe and nutritious food to meet dietary needs and preferences [20]. Experiences of household food insecurity vary depending on severity, ranging from anxiety about consistently accessing adequate food to compromising the quality and/or quantity of food consumed, which may involve hunger [21]. Household food insecurity has been associated with numerous adverse perinatal risk factors and outcomes in high-income countries [22,23,24,25,26,27,28,29,30]. The prevalence in Australia is not routinely monitored using suitably sensitive tools [31]; however, national prevalence estimates range from 4% [32] to 11.4% [33]. The household food insecurity prevalence among pregnant women is not currently known. 

An increased risk of household food insecurity is plausible in pregnancy, given the higher nutritional demands and potential for financial strain due to pregnancy-related expenses and reduced capacity to engage in paid work [34,35]. Additionally, it is well established that rates of food insecurity are higher in women [33,36]. Research indicates that mothers commonly compromise their own dietary intake to protect their children and other household members [37,38,39]. It is unclear whether household coping strategies change in pregnancy to protect the mother and foetus from the dietary impacts of food insecurity [40]; however, qualitative research indicates that some pregnant women prioritise their partner’s nutritional needs over their own [39].

The relationship between household food insecurity and diet quality has not been thoroughly examined among pregnant women [41,42,43]. Four studies conducted in the United States (US) have reported no association between household food insecurity and overall diet quality in pregnancy [35,40,44,45], in contrast to findings for non-pregnant adults in the US [42] and other high-income countries [41,42,43]. It has been suggested that this could be due to changes in food allocation within food insecure households to protect the pregnant woman [40,44] and/or participation in nutrition assistance programs for pregnant women with low incomes [45]. More research is therefore needed to understand the relationship between food insecurity and diet quality in pregnancy, particularly in other high-income countries where contextual differences, including responses to food insecurity, may influence findings.

The primary aim of this study was to examine the association between household food insecurity severity (marginal, low, and very low food security compared to high food security) and diet quality in pregnancy. It was hypothesised that food insecurity severity would be negatively associated with overall diet quality scores. Relationships between household food insecurity, dietary variety, and the odds of meeting fruit and vegetable recommendations (components of diet quality) were assessed as secondary aims, and the a priori hypotheses were that food insecurity would be associated with lower dietary variety and lower odds of meeting fruit and vegetable recommendations. 

## 2. Materials and Methods

### 2.1. Study Design and Participants

This cross-sectional study used survey methodology to collect self-reported data from participants. Eligible participants were currently pregnant, aged 16 years and above, resided in Australia, and proficient in English to allow for informed consent and participation in the online survey. For clarity [46], participants will be referred to as ‘pregnant women’, as although gender was not reported, the recruitment materials and methods targeted women and mothers. The authors acknowledge that not all pregnant people identify as women or mothers and affirm that all care should be respectful and responsive to individual needs and preferences. 

Ethics approval for this study was granted by the Metro North Human Research Ethics Committee A (protocol: HREC/2022/QRBW/82273; approved on 12 May 2022) and ratified by the Queensland University of Technology (QUT) and QIMR Berghofer Medical Research Institute Human Research Ethics Committees. All participants provided informed consent. 

This study is reported in accordance with the STROBE checklist for cross-sectional studies (Appendix A) [47] and the Checklist for Reporting Results of Internet E-Surveys (CHERRIES; Appendix A) [48].

### 2.2. Recruitment and Consent

Convenience sampling was used to recruit participants from antenatal clinics at three public hospitals in the Greater Brisbane region of Queensland, Australia, through display and/or distribution of flyers with a QR code. Participants were also recruited via social media advertising, including unpaid Facebook posts and paid Meta (Facebook and Instagram) advertising to pregnant women. Social media advertisements predominantly targeted potential participants residing in South East Queensland; however, given that snowball sampling can occur through user interactions with social media posts, recruitment was not limited to this region.

People living with disadvantage, who are more likely to experience food insecurity [19,20], are generally harder to reach and are frequently underrepresented in research [49]. Therefore, to ensure participants from a range of socioeconomic backgrounds were sufficiently represented in the sample, Meta (Facebook and Instagram) advertising was used to oversample lower socioeconomic status (SES) areas. The advertisements targeted women living in suburbs with lower Socioeconomic Indexes for Areas (SEIFA) [50] scores, which reflect higher levels of socioeconomic disadvantage and lower levels of advantage in the area. This oversampling was conducted over 1.5 months during the 6-month recruitment period. 

Eligible participants accessed the participant information online and were required to confirm their consent electronically before proceeding to the survey. Participants were advised prior to consenting that the estimated completion time was 25–35 min. As a gesture of appreciation, participants were offered entry in a prize draw to win one of three AUD 200 gift cards.

### 2.3. Data Collection and Integrity 

Data were collected between August 2022 and March 2023 via an online survey hosted by Qualtrics [51]. The survey included up to 244 items, with adaptive questioning used to skip irrelevant questions or follow-up items based on previous responses. 

Data integrity checks were completed to identify duplicate, suspicious, or poor-quality survey responses. Qualtrics’ bot detection (reCAPTCHA V3 scores < 0.5) and ‘speeder’ respondent identification (>2 SD from the median survey completion duration) features were used [52]. Additional data integrity checks included manual reviews of IP addresses, contact information (provided for prize draw entry and/or expression of interest to participate in other related research), open-ended responses, and survey completion times. Major inconsistencies in contact details, the use of temporary email addresses or uncommon email account providers, nonsensical or irrelevant open-ended responses, fast survey completion times, or exact matches for both start and finish times (unlikely to occur by chance) were considered suspicious or poor-quality [53,54] and formed the criteria for exclusion. Where duplicate responses were identified, the most complete and/or recent response was retained provided there were no other data integrity concerns.

Proportions of participants with a multiple pregnancy and participants who identified as Aboriginal and/or Torres Strait Islander were compared to the rates reported for pregnant women nationally (1.4% and 5%, respectively) [55], as significant variation (particularly higher proportions in the sample) may be indicative of poor data integrity. 

### 2.4. Measures

#### 2.4.1. Household Food Security Status

Household food security status was measured using the 10-item United States Department of Agriculture (USDA) Household Food Security Survey Module (HFSSM) [56]. The HFSSM items relate to financial access to adequate food over the past 12 months, and affirmative responses were summed to determine the severity of food insecurity among adults in the household as per the USDA protocol. Participants were categorised as having high food security (no affirmative responses), marginal food security (1–2 affirmative responses), low food security (3–5 affirmative responses), or very low food security (6–10 affirmative responses). 

Household food security status was also dichotomised into food secure (high food security) or food insecure households (marginal, low, or very low food security). Marginal food security was considered food insecure, as a growing body of research indicates that it is distinct from high food security and is associated with adverse health outcomes [57,58]. 

#### 2.4.2. Dietary Intake

Usual dietary intake over the past 6 months was assessed using a 107-item food frequency questionnaire (FFQ), with responses selected on a 9-point scale that ranged from ‘never or less than once per month’ to ‘6+ times per day’. This FFQ was used in the 1995 Australian National Nutrition Survey [59] and has since been modified and used in studies with Australian adults [60], including those of reproductive age [61]. The FFQ was further modified for this study to improve the suitability for a pregnant sample (removal of alcohol items) and to add three additional food items (muesli bars and other snack bars; fried chicken; and Chinese/Thai/Indian takeaway) that have been included in other Australian FFQs and studies [61,62]. The alcohol items were replaced by a single question from the Australian National Health Survey [63] that assessed frequency of alcohol intake, with the timeframe modified to refer to the current pregnancy. Alcohol frequency responses were collapsed to a dichotomous variable consisting of ‘no alcohol’ and ‘any alcohol consumed in pregnancy’. 

Dietary behaviours were also assessed via short questions from the 1995 Australian National Nutrition Survey [59], including valid questions about usual fruit and vegetable intake [64,65] and items about the use of salt, trimming of fat from meat, and the type of bread and milk typically consumed. 

#### 2.4.3. Diet Quality

Diet quality scores were derived from dietary intake data using an a priori index, the Dietary Guidelines Index 2013 (DGI-13) [60]. This index measures adherence to the Australian Dietary Guidelines [66] and has been used previously in pregnancy studies in Australia [67,68,69].

The DGI-13 [60] comprises 13 components, including both adequacy and moderation components, each scored from 0 to 10. The overall diet quality score was summed from these components (maximum possible score of 130), with higher scores reflecting greater adherence to dietary guidelines. The DGI-13 criteria [60] were modified for pregnancy (Appendix A) by altering the adequacy criteria to match the current recommendations for pregnancy [66,70] and modifying the alcohol component to reflect a binary score, with any prenatal alcohol consumption receiving the minimum score (0). 

DGI-13 subscores were calculated by combining responses for the FFQ items to estimate usual daily servings of core food groups (except for fruits and vegetables) and moderation components (unsaturated oils and spreads; discretionary foods; and added sugars). To calculate subscores for usual daily servings of fruits and vegetables, short dietary question responses were used as per previous applications of this index [60]. Responses for dietary behaviour questions were used to determine subscores for guidelines relating to limiting saturated fat intake and salt use, and choosing mostly wholegrains when consuming grain and cereal foods. 

#### 2.4.4. Dietary Variety

The variety component score from the DGI-13 [60] was used as the measure of dietary variety. This component score assesses the variety of vegetables, fruits, grains and cereal foods, dairy and dairy alternatives, and lean meat and meat alternatives consumed, using a similar approach as the Recommended Food Score [71]. Scores range from 0 to 10, with higher scores reflecting greater dietary variety. 

#### 2.4.5. Adherence to Fruit and Vegetable Recommendations 

Adherence to fruit and vegetable recommendations was determined by comparing usual daily servings of fruits and vegetables, assessed via valid short dietary questions [64,65], to the current recommendations [66]. Participants who indicated that they usually consumed at least 2 servings of fruit per day and at least 5 servings of vegetables per day were coded as meeting the recommendation. 

#### 2.4.6. Sociodemographic Variables

Self-reported sociodemographic characteristics were collected, including age in years, gestation in weeks, number of previous births, highest educational attainment, relationship status, gross annual household income range, and household composition. 

Equivalised household income is the household income adjusted for household size and composition. Equivalised household income was calculated by applying a modified OECD equivalence factor (derived from the number of adults and children reported in the household) to the median of the gross annual household income range selected by the participant [72]. Equivalised income was then collapsed into quintiles, and a dichotomous variable was created for lower household incomes, which was considered the lowest two quintiles [72]. Residential postcode was also collected to determine the SEIFA Index of Relative Socioeconomic Advantage and Disadvantage (IRSAD) decile [50]. Lower deciles reflect greater levels of socioeconomic disadvantage and lower levels of advantage in the area, as an indicator of lower area-level SES. 

### 2.5. Statistical Analyses

Data were first screened for missing values for the primary exposure and outcome variables. Missing data for HFSSM questions were imputed as per the USDA protocol [73]. This imputation procedure orders the items by their severity and applies coding rules based on whether affirmative and/or negative responses have been recorded prior to and following the missing item. Missing data for FFQ items were coded as ‘never or less than once per month’ if ≤10% of FFQ responses were missing. If >10% of responses were missing, data were considered invalid and the participant was excluded [60]. Missing data for sociodemographic variables were minimal (<4%). 

Sociodemographic and dietary variables were summarised by household food security status. Frequencies and proportions were used to describe categorical variables and mean and standard deviation (SD) were reported for continuous variables. Associations were tested via one-way ANOVA for continuous variables and ꭓ^2^ tests (or Fisher’s exact test if assumptions were violated) for categorical variables. 

Multiple linear regression models examined the association between food insecurity severity (reflected by marginal, low, and very low food security categories, with high food security as the reference) and total DGI-13 score (primary outcome) and DGI-13 dietary variety component score (secondary outcome). Binary logistic regression models were used to calculate unadjusted and adjusted odds ratios (OR) and 95% confidence intervals (CI) for associations between food insecurity and adherence to fruit and vegetable recommendations (secondary outcomes). A sensitivity analysis was conducted to confirm that the removal of outliers identified in the final model for the primary outcome did not significantly change the model fit or estimates.

To identify confounders, a directed acyclic graph [74] was used (see Appendix A). The development of the directed acyclic graph was informed by the literature and theoretical considerations. The confounding variables identified for adjustment in the regression models were maternal age (years; continuous), equivalised household income (lower household income; dichotomous), education (Bachelor’s degree or higher; dichotomous), and relationship status (married/de facto; dichotomous). These sociodemographic factors are associated with household food insecurity [19] and have consistently been reported as determinants of diet quality in pregnancy [17,18,75].

It was prespecified that a minimum sample of 210–300 participants was required for a multiple linear regression model with 7–10 independent variables [76]. Over-recruitment was performed to account for incomplete responses and to ensure the sample included sufficient representation of participants with a range of socioeconomic backgrounds. 

Analyses were conducted using IBM SPSS version 29.0 [77]. All statistical tests were two-sided and statistical significance was considered *p* < 0.05 for all analyses. 

## 3. Results

### 3.1. Participants

Of the 2220 eligible respondents who consented and commenced the online survey, 1540 (69.4%) provided sufficient data to determine their food security status and total diet quality score and were therefore included in this study (see Appendix A). A response rate could not be calculated due to the method of survey distribution. Compared to the participants included in this study, survey respondents who did not provide sufficient data for inclusion were significantly younger, less likely to hold a Bachelor’s degree or higher, less likely to have an annual household income of AUD 104,000 or higher, and more likely to live in a low SES area (Appendix A).

The characteristics of the sample and participants with high, marginal, low, and very low food security are presented in Table 1. The mean (SD, range) age was 30.6 (4.5, 17–41) years. A majority of participants resided in Queensland, were born in Australia, were university educated, and were either married or in a de facto relationship (Table 1). 

### 3.2. Household Food Insecurity

In the sample, 42% (*n* = 646) were food insecure, comprised of 19.4% (*n* = 299) reporting marginal food security, 11% (*n* = 169) reporting low food security, and 11.6% (*n* = 178) reporting very low food security among adults in the household. At least half (*n* = 92/173, 53.2%) of those who had experienced very low food security, where adults in the household had reduced the quantity of food consumed and may have gone hungry, had never used community food programs such as food banks, food parcels, and hampers. 

Compared to food secure participants (high food security), those who were food insecure (marginal, low, and very low food security) were, on average, younger by 2 years (95% CI: 1.5–2.4, t1228.10 = 8.36, *p* < 0.001), more likely to live in a lower-income household (OR: 5.71, 95% CI: 4.55–7.16, *p* < 0.001), and less likely to hold a Bachelor’s degree or higher (OR: 0.24, 95% CI: 0.19–0.30, *p* < 0.001). They were also more likely to have a child/children in the household (OR: 2.37, 95% CI: 1.93–2.92, *p* < 0.001), to have a pre-pregnancy body mass index (BMI) of 30 or above (OR: 2.1 95% CI: 1.7–2.7, *p* < 0.001), and to rate their health as fair or poor (OR: 3.28, 95% CI: 2.10–5.11, *p* < 0.001). 

### 3.3. Diet Quality

Overall, adherence to dietary guidelines was poor. No participants met all five pregnancy-specific adequacy (core food group) guidelines [66], and over one-third of participants (34.5%) did not meet any of these guidelines. However, the odds of not meeting any of the adequacy guidelines was higher for those with marginal (OR: 1.35, 95% CI: 1.02–1.78, *p* = 0.036), low (OR: 1.73, 95% CI: 1.24–2.43, *p* = 0.001) and very low food security (OR: 2.87, 95% CI: 2.07–3.99, *p* < 0.001) compared to those with high food security.

The total and component DGI-13 scores are shown in Table 1. The total DGI-13 scores ranged from a minimum of 39.8 to a maximum of 120.2 (maximum possible score: 130), with a mean (SD) of 76.2 (13.6). Compared to high food security, marginal, low, and very low food security were all significantly associated with poorer diet quality (Table 2). The DGI-13 scores decreased to a greater degree as food insecurity severity increased. All associations remained significant after adjusting for confounders; however, the estimates were attenuated to some extent (Table 2). Two outliers with standardised residuals >3 SD were identified in the adjusted model; however, a sensitivity analysis (Appendix A) indicated that they had a minimal influence on the model fit and beta-coefficients (0.4% change in adjusted *R*^2^; <5% change in estimates). 

### 3.4. Dietary Variety

Very low food security was significantly associated with lower dietary variety, with high food security as the reference (Table 3). This association remained significant after adjusting for age, education, equivalised household income, and relationship status, although there was some attenuation (Table 3). Associations between marginal and low food security and dietary variety were not statistically significant (Table 3). 

### 3.5. Fruit and Vegetable Recommendations

Few participants met the vegetable or both the fruit and vegetable intake recommendations, while approximately half met the fruit recommendation (Table 1). Compared to high food security, the odds of meeting the fruit recommendation were significantly lower with marginal (adjusted OR [AOR]: 0.73, 95% CI: 0.55–0.97, *p* = 0.028), low (AOR: 0.64, 95% CI: 0.45–0.92, *p* = 0.017), and very low food security (AOR: 0.41, 95% CI: 0.28–0.61, *p* < 0.001) after controlling for age, education, equivalised household income, and relationship status. Models were underpowered to assess associations between food insecurity severity and odds of meeting vegetable and both fruit and vegetable recommendations; therefore, food security status was dichotomised for subsequent modelling, and adjusting for education only produced the most parsimonious models (Appendix A). 

As shown in Table 4, food insecurity was associated with lower odds of meeting the fruit, vegetable, and both fruit and vegetable recommendations. After adjustment for education, the associations remained significant for lower odds of meeting the fruit and vegetable recommendations individually, but not for meeting both recommendations.

## 4. Discussion

The results of the present study indicate that all levels of food insecurity (including marginal food insecurity) were associated with poorer diet quality in pregnancy. This is consistent with previous findings among adults in general [41,42,43]; however, to the authors’ knowledge, this has not previously been reported in pregnancy in a high-income country context. These findings contrast with previous studies conducted in the US, which have reported no significant association between food insecurity and overall diet quality in pregnant populations [35,40,44,45]. This may be due to methodological differences in dietary assessment and indices used to assess diet quality, and/or contextual differences in the country- and region-specific responses to food insecurity and support available to those at risk in the perinatal period. For example, in the US, both ad hoc and systematic approaches may be available to assist pregnant women experiencing food insecurity including charitable food relief and, for those who are eligible, the federally funded Supplemental Nutrition Assistance Program (SNAP) and/or the nutrition program for Women, Infants and Children (WIC). These programs are designed to alleviate experiences of food insecurity and provide a safety net for those at nutritional risk, respectively [78], although uptake of WIC has been declining [79]. Similarly, in addition to charitable food relief, pregnant women with low incomes in the United Kingdom may receive assistance via the government-funded Healthy Start program, which increases access to healthy foods and nutritional supplements [80]. No such programs exist in Australia, where ad hoc access to charitable food relief remains the primary response to food insecurity [81,82]. This typically entails accessing food banks and/or community food programs that provide food parcels, hampers, or vouchers [82].

The decreases in diet quality related to food insecurity in the current study are clinically important. Marginal, low, and very low food security were associated with decreases in DGI-13 scores of 1.9, 3.6, and 5.3 points, respectively, after controlling for confounding variables. A decrease of up to five points in DGI scoring has previously been associated with the most severe form of food insecurity in a sample of young Australian adults [83]. A five-point lower DGI-13 score could be the equivalent of missing half of the recommended daily fruit or vegetable servings, for example, while a decrease of approximately two points could be equivalent to consuming at least three fewer daily servings of grain and cereal foods, many of which are major sources of folic acid, iodine, and iron (micronutrients of particular importance in pregnancy) in the Australian food supply [84]. 

Limited adherence to fruit and vegetable recommendations has been consistently reported among pregnant women in Australia [12,13,85]; however, the present study provides evidence for an association between household food insecurity and poorer adherence. Food insecurity has previously been associated with lower fruit and vegetable intakes among pregnant women in the US [86]. 

Inadequate fruit and vegetable intakes in pregnancy have been linked to lower dietary fibre intakes [87], which may contribute to constipation, a common symptom in pregnancy [88], and could have implications for gestational weight gain [89]. A low dietary fibre intake can also influence the gut microbiome [90], which is increasingly implicated in metabolic, immunological, and neurological programming in early life [91,92]. 

Higher dietary variety, which increases the likelihood of meeting nutritional requirements [66], is also associated with differences in the gut microbiome [93,94]. Some restriction to dietary variety may occur when the quality of food consumed is compromised to cope with low food security and very low food security [21]. However, in this sample, a decrease in dietary variety was only associated with very low food security (the most severe level of household food insecurity, which also impacts the quantity of food consumed).

The results of this Australian survey suggest that household food insecurity is prevalent among pregnant women. Relative to pregnant women nationally, participants of this study may be more advantaged overall. The mean age of participants was similar to the national average maternal age at birth (30.6 years and 31.1 years, respectively) [55]. However, a higher proportion of participants were born in Australia (82.5% versus 66%), and a lower proportion identified as Aboriginal and/or Torres Strait Islander (2.9% versus 5%) [55]. Additionally, the current sample had a higher proportion of participants with a Bachelor’s degree or higher (approximately 60% compared to 50% of Australian women aged 25–44 years) [95] and higher rates of participants accessing private care for their pregnancy (28.8% versus 25.4% of pregnant women nationally) [55]. Therefore, the prevalence of food insecurity could be even higher in the pregnant population nationally. 

The current pregnancy care guidelines in Australia [88] recommend that women receive advice about a healthy diet, in accordance with the Australian Dietary Guidelines [66], at each antenatal visit. While pregnancy is often referred to as a ‘teachable moment’ [96], awareness of the importance of a healthy diet in pregnancy and the inability to access this in the context of food insecurity could potentially contribute to higher psychological stress [96,97], which has been associated with an increased risk of adverse perinatal outcomes [98]. The provision of dietary advice in accordance with the national guidelines was not assessed in the current study; however, approximately half of the participants who were food insecure recalled receiving some form of dietary advice from a health professional in their pregnancy so far. This suggests that a substantial proportion of women may receive prenatal dietary advice that they are unable to implement. 

The findings of this study suggest that consistent access to enough nutritious food is out of reach for many Australian families in pregnancy, which raises significant concerns for maternal and child health. This underscores the need for a systematic approach to identify and assist those who are at risk of food insecurity. One strategy may be the implementation of routine screening in pregnancy care to enable referrals to appropriate support services [99]. In this study, over half of the participants who had experienced very low food security had not accessed community food assistance, such as food banks and food parcels or hampers. While the current study did not explore the reasons for this, research indicates that accessing these services is often stigmatising and practical barriers are common, such as long queues and waiting times [82,100]. Several concerns have also been highlighted with the quality and appropriateness of food provided by these services, including a lack of familiar and/or culturally appropriate foods, a lack of nutritious foods, difficulties accessing foods that meet specific dietary requirements (such as food allergies), and the provision of food that is out of date [81,82]. There is a clear need to understand what place-based policy and service system changes, and community-health service partnerships, are required to provide those who are at risk of food insecurity with non-stigmatising, suitable support. 

### Strengths and Limitations

The strengths of this study include the large sample size and use of a validated tool for the measurement of food insecurity rates and severity [20]. The diet quality index used also encompassed all four dimensions of diet quality, namely adequacy, variety, balance, and moderation [101], and measured adherence to the dietary guidelines referenced in the current Australian clinical practice guidelines for pregnancy care [88].

There are several limitations of the present study, including its cross-sectional design, which limits the inference of causality. Due to the timeframe assessed using the HFSSM (the last 12 months), some food secure participants may potentially experience food insecurity later on in their pregnancies (after completion of the survey), and some food insecure participants may have experienced food insecurity in the pre-conception period rather than during pregnancy. Nevertheless, pre-conception diet and health influence pregnancy and child health [102]; thus, food insecurity during this time is also concerning. 

The measurement of diet quality in the current sample is also subject to limitations. The FFQ and DGI-13 have not been validated specifically for use in pregnancy, although they have previously been used among Australian adults of reproductive age and with pregnant samples [61,67,69,103]. Furthermore, in accordance with the protocol used to develop the DGI-13 and previous applications of this index [60], assumed serving sizes were applied for the conversion of qualitative FFQ responses to DGI-13 scoring; therefore, some measurement error is likely. The FFQs also have other limitations, including the potential for social desirability bias and recall error [104]; responses may potentially be biased by more recent intakes [59]. An FFQ was used in this study to better align with the timeframe in which food insecurity was assessed. Food insecurity can have a cyclical nature, with the severity changing across pay or entitlement cycles [105]. This can contribute to feast–famine cycles [106]; therefore, short-term measures such as a 24 h recall may potentially capture dietary intake at a time when food access is adequate or improved. 

Although the authors attempted to identify and control for confounding factors, there is potential for residual confounding from unmeasured variables, such as the level of nutrition knowledge. Holding a Bachelor’s degree or higher has been associated with higher levels of nutrition knowledge in Australia [11], and this was controlled for in all adjusted regression models. Pregnancy-related symptoms, such as nausea and vomiting, were also not assessed. Previous research suggests that diet quality does not change significantly across trimesters [107] despite the prevalence of nausea and vomiting differing from early to late pregnancy [108]. Distributions of participants by trimester of pregnancy were similar across all food security status categories in the current study. 

Finally, completion of the online survey required a level of literacy and time availability, which may have precluded participation for some [49]. While there was reasonable representation across all IRSAD deciles in the sample, this is not an indicator of SES at the individual or household level and it is unlikely that those living with the greatest levels of socioeconomic disadvantage were recruited. Most participants were also recruited via social media, which may have biased the sample, and it is acknowledged that this study’s advertisements targeted women and mothers; therefore, the results may not be generalisable to all pregnant people. 

## 5. Conclusions

This survey undertaken in Australia demonstrated that household food insecurity is associated with decreased diet quality in pregnancy. As food insecurity severity increased, the overall diet quality scores decreased to a greater extent. Food insecurity was prevalent in the sample, suggesting that many families are impacted during this critical life stage. The implications of food insecurity in pregnancy are significant, including an increased risk of adverse perinatal outcomes and the potential for long-lasting impacts on maternal and child health. This issue therefore requires urgent attention to ensure that all families can consistently access the material resources they need, including access to enough nutritious food in socially acceptable ways, to support health in pregnancy and across the life course.

## Figures and Tables

**Table 1 nutrients-16-01319-t001:** Characteristics of a sample of 1540 pregnant women in Australia.

	Mean (SD ^a^) or *n* (%)					
	All Participants (*n* = 1540)	High Food Security (*n* = 894)	Marginal Food Security (*n* = 299)	Low Food Security (*n* = 169)	Very Low Food Security (*n* = 178)	*p* Value ^b^
Age (years)	30.6 (4.5)	31.4 (4.0)	30.2 (4.7)	29.8 (4.8)	27.9 (4.9)	<0.001
Trimester						0.266
First	307 (19.9)	170 (19.0)	61 (20.4)	42 (24.9)	34 (19.1)	
Second	703 (45.6)	411 (46.0)	135 (45.2)	65 (38.5)	92 (51.7)	
Third	530 (34.4)	313 (35.0)	103 (34.4)	62 (36.7)	52 (29.2)	
Pregnancy type						0.283
Singleton	1516 (98.4)	882 (98.7)	296 (99.0)	164 (97.0)	174 (97.8)	
Twins	24 (1.6)	12 (1.3)	3 (1.0)	5 (3.0)	4 (2.2)	
Pregnancy care						<0.001
Public	1073 (71.2)	549 (62.7)	229 (78.4)	136 (81.4)	159 (91.9)	
Private	435 (28.8)	327 (37.3)	63 (21.6)	31 (18.6)	14 (8.1)	
Missing	32	18	7	2	5	
Previous births						<0.001
0	855 (55.5)	568 (63.5)	150 (50.2)	68 (40.2)	69 (38.8)	
1	462 (30.0)	257 (28.7)	89 (29.8)	63 (37.3)	53 (29.8)	
>1	223 (14.5)	69 (7.7)	60 (20.1)	38 (22.5)	56 (31.5)	
Children in household						<0.001
No	817 (53.1)	554 (62.0)	141 (47.2)	64 (37.9)	58 (32.6)	
Yes	723 (46.9)	340 (38.0)	158 (52.8)	105 (62.1)	120 (67.4)	
Residential state						0.023
Queensland	1495 (97.1)	866 (96.9)	285 (95.3)	166 (98.2)	178 (100)	
Other	45 (2.9)	28 (3.1)	14 (4.7)	3 (1.8)	0 (0)	
Born in Australia						0.008
No	267 (17.5)	175 (19.8)	51 (17.1)	24 (14.5)	17 (9.6)	
Yes	1260 (82.5)	710 (80.2)	248 (82.9)	142 (85.5)	160 (90.4)	
Missing	13	9	0	3	1	
Aboriginal and/or Torres Strait Islander						<0.001
No	1495 (97.1)	877 (98.1)	292 (97.7)	163 (96.4)	163 (91.6)	
Yes	45 (2.9)	17 (1.9)	7 (2.3)	6 (3.6)	15 (8.4)	
Married/de facto						<0.001
No	105 (6.9)	24 (2.7)	22 (7.5)	17 (10.2)	42 (23.9)	
Yes	1427 (93.1)	870 (97.3)	273 (92.5)	150 (89.8)	134 (76.1)	
Missing	8	0	4	2	2	
Bachelor’s degree or higher						<0.001
No	621 (40.4)	235 (26.3)	136 (45.5)	100 (59.5)	150 (84.3)	
Yes	918 (59.6)	659 (73.7)	163 (54.5)	68 (40.5)	28 (15.7)	
Missing	1	0	0	1	0	
Household income (AUD ^c^)						<0.001
0–25,999	54 (3.6)	6 (0.7)	5 (1.7)	15 (9.2)	28 (16.3)	
26,000–51,999	110 (7.3)	23 (2.6)	26 (9.0)	21 (12.9)	40 (23.3)	
52,000–103,999	415 (27.7)	165 (18.9)	95 (33.0)	77 (47.2)	78 (45.3)	
104,000–207,999	705 (47.1)	491 (56.1)	143 (49.7)	45 (27.6)	26 (15.1)	
208,000 or higher	214 (14.3)	190 (21.7)	19 (6.6)	5 (3.1)	0 (0)	
Missing	42	19	11	6	6	
Equivalised ^d^ household income						<0.001
Quintile 1	306 (20.4)	66 (7.5)	69 (24.0)	66 (40.5)	105 (61.0)	
Quintile 2	299 (20.0)	141 (16.1)	66 (22.9)	49 (30.1)	43 (25.0)	
Quintile 3	361 (24.1)	237 (27.1)	82 (28.5)	28 (17.2)	14 (8.1)	
Quintile 4	351 (23.4)	265 (30.3)	59 (20.5)	17 (10.4)	10 (5.8)	
Quintile 5	181 (12.1)	166 (19.0)	12 (4.2)	3 (1.8)	0 (0)	
Missing	42	19	11	6	6	
Lower household income ^e^						<0.001
No	893 (59.6)	668 (76.3)	153 (53.1)	48 (29.4)	24 (14.0)	
Yes	605 (40.4)	207 (23.7)	135 (46.9)	115 (70.6)	148 (86.0)	
Missing	42	19	11	6	6	
SEIFA-IRSAD ^f^ score (deciles)						<0.001
Low (1–3)	319 (20.7)	137 (15.4)	61 (20.5)	48 (28.6)	73 (41.2)	
Medium (4–7)	584 (37.9)	311 (34.9)	133 (44.6)	72 (42.9)	68 (38.4)	
High (8–10)	632 (41.0)	444 (49.8)	104 (34.9)	48 (28.6)	36 (20.3)	
Missing	5	2	1	1	1	
Smoking status						<0.001
Non-smoker	1503 (97.6)	887 (99.2)	292 (97.7)	164 (97.0)	160 (89.9)	
Smoker (any frequency)	37 (2.4)	7 (0.8)	7 (2.3)	5 (3.0)	18 (10.1)	
Pre-pregnancy BMI ^g^ category						<0.001
<18.5	52 (3.5)	29 (3.3)	8 (2.8)	4 (2.5)	11 (6.6)	
18.5–24.9	720 (48.6)	476 (54.7)	132 (45.8)	55 (35.0)	57 (34.3)	
25.0–29.9	372 (25.1)	215 (24.7)	81 (28.1)	41 (26.1)	35 (21.1)	
≥30.0	337 (22.8)	150 (17.2)	67 (23.3)	57 (36.3)	63 (38.0)	
Missing	59	24	11	12	12	
Self-reported health						<0.001
Poor or fair	96 (6.2)	30 (3.4)	25 (8.4)	12 (7.1)	29 (16.3)	
Good	555 (36.0)	250 (28.0)	122 (40.8)	82 (48.5)	101 (56.7)	
Very good or excellent	889 (57.7)	614 (68.7)	152 (50.8)	75 (44.4)	48 (27.0)	
Received dietary advice from a health professional in this pregnancy so far						<0.001
No	665 (44.5)	352 (40.2)	149 (52.1)	78 (48.4)	86 (50.3)	
Yes	829 (55.5)	524 (59.8)	137 (47.9)	83 (51.6)	85 (49.7)	
Missing	46	18	13	8	7	
Met fruit recommendation (≥2 servings/day)	859 (55.8)	561 (62.8)	157 (52.5)	80 (47.3)	61 (34.3)	<0.001
Met vegetable recommendation (≥5 servings/day)	53 (3.4)	44 (4.9)	7 (2.3)	0 (0)	2 (1.1)	0.001
Met fruit and vegetable recommendations	43 (2.8)	35 (3.9)	6 (2.0)	0 (0)	2 (1.1)	0.005
DGI-13 ^h^ component score ^i^ (max. score)						
Dietary variety (10)	4.6 (1.4)	4.7 (1.4)	4.6 (1.4)	4.5 (1.5)	3.9 (1.5)	<0.001
Vegetables (10)	4.2 (2.3)	4.7 (2.3)	4.1 (2.2)	3.3 (1.9)	2.9 (1.9)	<0.001
Fruits (10)	7.4 (3.1)	7.9 (2.9)	7.1 (3.2)	7.0 (3.0)	5.6 (3.5)	<0.001
Grains and cereals (10)	4.0 (2.6)	4.3 (2.5)	4.0 (2.7)	3.5 (2.7)	2.8 (2.5)	<0.001
Meat and alternatives (10)	7.2 (1.6)	7.3 (1.5)	7.2 (1.6)	7.0 (1.6)	6.4 (1.7)	<0.001
Dairy and alternatives (10)	5.2 (2.9)	5.2 (2.8)	5.4 (3.0)	5.4 (2.9)	4.9 (3.1)	0.22
Fluids (10)	8.4 (2.0)	8.6 (1.8)	8.2 (2.2)	8.1 (2.2)	7.7 (2.4)	<0.001
Limit discretionary foods (10)	3.2 (4.7)	3.3 (4.7)	2.9 (4.6)	2.7 (4.4)	3.1 (4.6)	0.239
Limit saturated fats (10)	5.6 (3.5)	5.9 (3.4)	5.5 (3.6)	5.1 (3.6)	4.7 (3.5)	<0.001
Limit unsaturated fats (10)	9.6 (1.9)	9.7 (1.8)	9.5 (2.2)	9.5 (2.1)	9.6 (2.1)	0.511
Limit added salt (10)	4.3 (2.9)	4.5 (2.9)	4.1 (3.0)	4.2 (2.9)	4.2 (2.9)	0.155
Limit added sugars (10)	3.2 (4.7)	3.2 (4.7)	3.2 (4.7)	3.0 (4.6)	3.7 (4.8)	0.564
No alcohol (10)	9.4 (2.4)	9.3 (2.6)	9.4 (2.3)	9.6 (2.0)	9.6 (2.1)	0.18
DGI-13 ^h^ total score ^i^ (max. score: 130)	76.2 (13.6)	78.6 (13.1)	75.1 (12.8)	72.9 (13.8)	68.7 (13.7)	<0.001

^a^ SD: standard deviation. ^b^ *p* values derived from one-way ANOVAs for continuous variables and ꭓ^2^ tests or Fisher’s exact test (if assumptions for ꭓ^2^ test were violated) for categorical variables. ^c^ AUD: Australian dollars. ^d^ Annual gross household income adjusted for household size and composition using a modified OECD equivalence factor. ^e^ Equivalised household income quintiles 1–2. ^f^ SEIFA-IRSAD: Socioeconomic Indexes for Areas–Index of Relative Socioeconomic Advantage and Disadvantage. Deciles were derived from self-reported residential postcode as indicators of area-level socioeconomic status. Lower deciles reflect higher relative socioeconomic disadvantage and lower relative advantage in the area (vice versa for higher deciles). ^g^ BMI: body mass index. ^h^ DGI-13: Dietary Guidelines Index 2013. ^i^ Higher scores reflect greater adherence to dietary guideline(s).

**Table 2 nutrients-16-01319-t002:** Associations of household food insecurity severity with diet quality, as measured by total Dietary Guidelines Index 2013 (DGI-13) score, in a sample of 1540 pregnant women in Australia.

	Model 1 (*n* = 1540)		Model 2 (*n* = 1492) ^a^	
	Adjusted *R*^2^: 6%		Adjusted *R*^2^: 9.9%	
	Unadjusted β (95% CI ^b^)	*p* Value	Adjusted ^a^ β (95% CI ^b^)	*p* Value
High food security	Reference		Reference	
Marginal food security	−3.5 (−5.3, −1.8)	<0.001	−1.9 (−3.7, −0.1)	0.036
Low food security	−5.8 (−7.9, −3.6)	<0.001	−3.6 (−5.9, −1.3)	0.002
Very low food security	−9.9 (−12.1, −7.8)	<0.001	−5.3 (−7.7, −2.8)	<0.001

^a^ Adjusted for age, education, equivalised household income, and relationship status; *n* for adjusted model lower due to missing data for covariates. ^b^ CI: confidence interval.

**Table 3 nutrients-16-01319-t003:** Associations of household food insecurity severity with dietary variety, as measured by the Dietary Guideline Index 2013 (DGI-13) variety component score, in a sample of 1540 pregnant women in Australia.

	Model 1 (*n* = 1540)		Model 2 (*n* = 1492) ^a^	
	Adjusted *R*^2^: 3.4%		Adjusted *R*^2^: 6.1%	
	Unadjusted β (95% CI ^b^)	*p* Value	Adjusted ^a^ β (95% CI ^b^)	*p* Value
High food security	Reference		Reference	
Marginal food security	−0.14 (−0.33, 0.04)	0.130	−0.01 (−0.20, 0.18)	0.887
Low food security	−0.19 (−0.42, 0.04)	0.106	0.01 (−0.23, 0.26)	0.917
Very low food security	−0.87 (−1.10, −0.64)	<0.001	−0.47 (−0.73, −0.21)	<0.001

^a^ Adjusted for age, education, equivalised household income, and relationship status; *n* for adjusted model lower due to missing data for covariates. ^b^ CI: confidence interval.

**Table 4 nutrients-16-01319-t004:** Associations of household food insecurity ^a^ with odds of meeting fruit and vegetable intake recommendations in a sample of 1540 pregnant women in Australia.

	Model 1 (*n* = 1540)		Model 2 (*n* = 1539) ^a,b^	
	OR ^c^ (95% CI ^d^)	*p* Value	AOR ^b,e^ (95% CI ^d^)	*p* Value
Meeting fruit recommendation	0.51 (0.41–0.62)	<0.001	0.61 (0.49–0.76)	<0.001
Meeting vegetable recommendation	0.27 (0.13–0.56)	<0.001	0.40 (0.19–0.84)	0.016
Meeting fruit and vegetable recommendations	0.31 (0.14–0.67)	0.003	0.45 (0.20–1.00)	0.051

^a^ The reference group is high food security (food secure). Food security status was dichotomised (marginal, low, and very low food security collapsed to form the food insecure group) due to low cases meeting the vegetable and fruit and vegetable recommendations, which limited power. ^b^ Adjusted for education. Modelling indicated that this produced the most parsimonious models (Appendix A). *n* is lower due to missing data for education (*n* = 1). ^c^ OR: odds ratio (unadjusted). ^d^ CI: confidence interval. ^e^ AOR: adjusted odds ratio.

## Data Availability

The data presented in this study are available upon request from the corresponding author. The data are not publicly available due to privacy.

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
