# Peer review of "Food Insecurity Is Associated with Diet Quality in Pregnancy: A Cross-Sectional Study"

_nutrients, 2024, doi:10.3390/nu16091319_

Round 1
Reviewer 1 Report
Comments and Suggestions for Authors
First of all, I would like to congratulate the authors for their research, which is of considerable interest and provides evidence on a subject that has been little explored in developed countries.
I would like to make some considerations and suggestions to improve the quality and coherence of their work.
In the present study, different aspects of diet (dietary intake, diet quality, dietary variety, adherence to fruit and vegetable recommendations) are associated with household food insecurity as well as with certain socioeconomic variables. Therefore, the title does not seem appropriate to me as it does not accurately reflect the content.
I suggest you:
Food insecurity and diet in Australian pregnant women. A cross-sectional study
The objective should also be expressed more clearly: to explore the association between food insecurity, socio-demographic variables and various dietary characteristics (Dietary intake, Diet quality, Dietary variety, Adherence to fruit and vegetable recommendations).
The nomenclature in the tables should be unified. Tables 2 and 3 show beta coefficients, while Table 4 shows OR. Likewise, in table 2 the coefficients of determination (R2) are shown while in tables 3 and 4 they are not.
I suggest that the conclusions section be more concrete, incorporating some fundamental numerical results.
Reviewer 2 Report
Comments and Suggestions for Authors
The authors used a cross-sectional online survey to investigate the relationship between household food insecurity and diet quality in pregnancy. The paper reported that food insecurity was found in almost all samples and the implication of food insecurity were significant in pregnancy. The results were interesting and useful.
1. The introduction was too tedious, please shorten this part.
2. The presentation of the results should be various. In the paper, the authors used a lot of tables to show their findings. The application of figures can make the data more visualized.
3. Please shorten the “Strengths and limitations” section. The advantages and disadvantages should be presented in this part. And the discussion part can be moved to the above section “Discussion”.
4. The format of references should be checked and unified according to the guideline of Nutrients.
